# EEG Changes during Propofol Anesthesia Induction in Vegetative State Patients Undergoing Spinal Cord Stimulation Implantation Surgery

**DOI:** 10.3390/brainsci13111608

**Published:** 2023-11-20

**Authors:** Xuewei Qin, Xuanling Chen, Bo Wang, Xin Zhao, Yi Tang, Lan Yao, Zhenhu Liang, Jianghong He, Xiaoli Li

**Affiliations:** 1Department of Anesthesiology, Peking University International Hospital, Beijing 102206, China; qinxuewei1999@163.com (X.Q.); zhaoxin@pkuih.edu.cn (X.Z.);; 2Institute of Electrical Engineering, Yanshan University, Qinhuangdao 066004, China; zhl@ysu.edu.cn; 3Department of Neurosurgery, Beijing Tiantan Hospital, Capital Medical University, Beijing 100070, China; he_jianghong@sina.cn; 4State Key Laboratory of Cognitive Neuroscience and Learning, IDG/McGovern Institute for Brain Research, Beijing Normal University, Beijing 100875, China; xiaoli@bnu.edu.cn; 5Center for Collaboration and Innovation in Brain and Learning Sciences, Beijing Normal University, Beijing 100875, China

**Keywords:** vegetative state, propofol, bispectral index, electroencephalogram, time–frequency spectrum, power spectrum

## Abstract

Objective: To compare the EEG changes in vegetative state (VS) patients and non-craniotomy, non-vegetative state (NVS) patients during general anesthesia with low-dose propofol and to find whether it affects the arousal rate of VS patients. Methods: Seven vegetative state patients (VS group: five with traumatic brain injury, two with ischemic–hypoxic VS) and five non-craniotomy, non-vegetative state patients (NVS group) treated in the Department of Neurosurgery, Peking University International Hospital from January to May 2022 were selected. All patients were induced with 0.5 mg/kg propofol, and the Bispectral Index (BIS) changes within 5 min after administration were observed. Raw EEG signals and perioperative EEG signals were collected and analyzed using EEGLAB in the MATLAB software environment, time–frequency spectrums were calculated, and EEG changes were analyzed using power spectrums. Results: There was no significant difference in the general data before surgery between the two groups (*p* > 0.05); the BIS reduction in the VS group was significantly greater than that in the NVS group at 1 min, 2 min, 3 min, 4 min, and 5 min after 0.5 mg/kg propofol induction (*p* < 0.05). Time–frequency spectrum analysis showed the following: prominent α band energy around 10 Hz and decreased high-frequency energy in the NVS group, decreased high-frequency energy and main energy concentrated below 10 Hz in traumatic brain injury VS patients, higher energy in the 10–20 Hz band in ischemic–hypoxic VS patients. The power spectrum showed that the brain electrical energy of the NVS group was weakened R5 min after anesthesia induction compared with 5 min before induction, mainly concentrated in the small wave peak after 10 Hz, i.e., the α band peak; the energy of traumatic brain injury VS patients was weakened after anesthesia induction, but no α band peak appeared; and in ischemic–hypoxic VS patients, there was no significant change in low-frequency energy after anesthesia induction, high-frequency energy was significantly weakened, and a clear α band peak appeared slightly after 10 Hz. Three months after the operation, follow-up visits were made to the VS group patients who had undergone SCS surgery. One patient with traumatic brain injury VS was diagnosed with MCS-, one patient with ischemic–hypoxic VS had increased their CRS-R score by 1 point, and the remaining five patients had no change in their CRS scores. Conclusions: Low doses of propofol cause great differences in the EEG of different types of VS patients, which may be the unique response of damaged nerve cell residual function to propofol, and these weak responses may also be the basis of brain recovery

## 1. Introduction 

A vegetative state (VS) is a pathological brain state that falls under the category of chronic brain disorders. Consciousness disorders are characterized by a clinical state wherein individuals exhibit a complete lack of self-awareness and environmental perception while still maintaining sleep–wake cycles and some level of hypothalamic and brainstem autonomic functions [1]. At present, there is a lack of specific methods for regaining consciousness of VS patients in clinical practice, and conservative treatment with conventional drugs has little effect [2,3]. Spinal Cord Stimulation (SCS) has been shown to improve the level of consciousness in VS patients [4,5]. However, according to the requirements of the surgeon, the patient needs to be in a prone position and remain motionless to perform the operation, and such patients cannot cooperate, so general anesthesia is the only option for them. Propofol is a short-acting anesthetic and sedative agent mainly acting on gamma-aminobutyrate (GABA) receptors [6], which can enhance the synaptic inhibition mediated by cortical GABA receptors and inhibit cerebral cortex discharge [7,8,9] to achieve sedation. Because it can reduce the brain’s metabolic rate [10], inhibit intracellular calcium release [11], and play an antioxidant role [12], it has a brain-protective effect and was widely used in the anesthesia of neurosurgery patients. Over the past few years, EEG-related research has made tremendous progress in predicting and detecting consciousness recovery after severe brain injury. But for VS patients with different types of injury, does propofol have any effect on the injured brain when induced? The pattern of the brain’s spectral response is not clear. Some studies have pointed out that high-dose propofol may aggravate severe craniocerebral injury and neurological dysfunction [13], but low-dose propofol can cause excitation-related changes in brain electricity and behavior [14,15]. The main purpose of this study was to use low-dose propofol to maintain surgical anesthesia and explore whether changes in EEG could contribute to better outcomes for VS patients while reducing brain injury. From January to May 2022, our Department of Neurosurgery treated seven patients in vegetative states who underwent SCS-implantation surgery. All of them used the BIS [16] (BIS refers to the determination of linear components of EEG (frequency and power), analysis of the nonlinear relationship between component waves (phase and harmonic), the selection of various EEG signals representing different sedation levels, standardization, and digital processing, which is a more accurate method to judge the sedation level and monitor the depth of anesthesia. It calculates a single dimensionless variable between 0 and 100. A patient with an isoelectric EEG has a score of 0, while, if fully awake, should have a BIS score close to 100. BIS scores of 40 to 55 are typical during general anesthesia) to monitor the EEG changes of such patients under low-dose propofol-induced anesthesia. The report is as follows.

## 2. Materials and Methods

### 2.1. General Information

Seven vegetative state patients (VS group) who underwent SCS implantation surgery and five non-craniotomy, non-vegetative state patients (NVS group) who underwent general anesthesia at Peking University International Hospital from January to May 2022 were selected as research subjects. General information on the VS group and the NVS group is shown in Table 1 and Table 2. Among the seven patients comprising the vegetative state (VS) group, five individuals exhibited VS as a result of traumatic brain injury, while the remaining two patients experienced VS due to hypoxic injury resulting from cardiac arrest. The duration between injury and SCS implantation in the VS group patients varied from 2 to 9 months, with an average duration of 5.43 ± 2.23 months. Prior to the surgical procedure, all three patients had undergone more than one month of hyperbaric oxygen chamber treatment, acupuncture, arousal drugs, and other regular comprehensive arousal-rehabilitation treatments, all of which proved ineffective in arousing the patients. Additionally, all patients met the criteria for SCS implantation surgery. All seven VS patients had undergone tracheotomy and had a tracheotomy tube in place, and all were diagnosed with pulmonary infection.

### 2.2. Anesthesia Method and Perioperative Management

All patients underwent continuous monitoring of electrocardiogram (ECG), peripheral oxygen saturation (SPO_2_), body temperature, invasive radial artery pressure, and BIS. Following the establishment of invasive arterial access, arterial blood samples were collected for blood gas analysis. The VS group received a breathing circuit connected to the tracheostomy tube in spontaneous ventilation mode. Anesthesia was induced using a dose of 0.5 mg/kg of propofol, and changes in BIS were observed at 1, 2, 3, 4, and 5 min after administration. After 5 min, intravenous injections of 0.3 μg/kg of sufentanil and 0.5 mg/kg of rocuronium were administered, and mechanical ventilation was initiated after spontaneous breathing ceased. NVS group patients were given oxygen by mask, 0.5 mg/kg propofol was given and BIS changes were observed at 1 min, 2 min, 3 min, 4 min, and 5 min after administration. After 5 min, 2 mg/kg of propofol, 0.3 ug/kg of sufentanil, and 0.5 mg/kg of rocuronium were given again, and laryngeal mask ventilation was started when BIS reached 40–50. Both groups of patients maintained PetCO_2_ between 35 ad 45 mmHg, anesthesia maintenance was performed with 1.5 mg/kg·h of propofol and 0.15 μg/kg·min of remifentanil infused via a micro-pump, and the dose was fine-tuned according to BIS to keep it within the normal range for routine patients. If the BIS in the Non-Volatile Substance (NVS) group surpassed 60, the propofol infusion dosage was promptly escalated to ensure the BIS remained within the range of 40–60. In the event that the intraoperative systolic pressure dropped by 10–20% compared to the baseline value, ephedrine in the dosage of 3–6 mg was immediately administered to elevate blood pressure. Additionally, if necessary, a continuous infusion of norepinephrine was employed. To maintain the acid–base balance and electrolyte levels within the normal range, appropriate corrective measures were undertaken based on the results of intraoperative blood gas analysis. After the operation, propofol and remifentanil infusion was stopped; after the NVS group patients regained consciousness and spontaneous breathing was restored, the laryngeal mask was removed, and they were transferred to the recovery room for further observation. They were returned to the ward after their vital signs were stable and they met the criteria for leaving the room; VS-group patients were returned to the ward after they regained spontaneous breathing, their SPO_2_ with room air was close to preoperative status, circulation was stable, they were observed in the recovery room, and they were returned to the ward after their vital signs were stable and they met the criteria for leaving the room.

### 2.3. Spinal Electrode Implantation Method

VS-group patients were placed in the prone position. The surgeon determined the T_7_-T_8_ interspace under X-ray guidance, the puncture needle entered at a 45° angle to the skin, the electrode was placed into the epidural space under X-ray guidance, the upper edge of the electrode was leveled with the middle edge of the C_2_ vertebrae, the epidural was in the middle, the impedance was tested to be normal, and the electrode was fixed.

### 2.4. Observation Indicators and Data Collection

The EEG of the two groups of patients was collected from the time of room entry. A total of 0.5 mg/kg of propofol induction was administered until the end of the operation by monitoring BIS, and time–frequency spectrum analysis of the EEG of the two groups of patients was performed to observe the pattern of EEG changes; the CRS-R scale was assessed by the same neurosurgeon before surgery and 3 months after surgery.

### 2.5. EEG Data Collection and Preprocessing

In this study, BIS EEG monitor from Medtronic was used to record dual-channel frontal EEG signals from patients at a sampling rate of 128 Hz. We wiped the patient’s forehead skin with an alcoholic cotton ball to ensure that the impedance of each channel electrode was kept to less than 5 kΩ. We arranged the electrodes according to FPz-F9 and FPz-AF7 of the international 10–20 system, which covered the brain in the same hemisphere. In this study, we only performed EEG signal analysis on the FPz-F9 single channel.

EEGLAB toolbox in MATLAB software environment is used to perform the preprocessing process, which takes into account both interference signals and signal integrity. The preprocessing steps are as follows:(1)Removal of large-amplitude noise. Loop noise is likely to occur at points of abrupt data changes, and large-amplitude noise will also appear at the marked points in the EEG signal from the BIS monitor. These large noises are first removed (the data in their time period are visually manually excluded). Specifically, the collected data are imported into the EEGLAB toolbox, and each patient’s data are marked out using the eegplot.m function with a time window of 100 and an amplitude of 200 μV. The noise is visually selected, and the large noise data in that segment are deleted.(2)Eliminate baseline drift and head movement artifacts. The POP_eEGFiltNews.m function of EEGLAB toolbox is used to reduce the low-frequency components below 0.1 Hz in EEG signals, and the effect of baseline drift is reduced by bidirectional least square method or window function FIR filter.(3)Remove power frequency interference. In order to reduce AC interference, an adaptive notch filter is used to process signals in 50 Hz and its narrow neighborhood.(4)The Clean_rawdata function of EEGLAB is used to delete bad data parts. Firstly, the Artifact Subspace Reconstruction (ASR) algorithm is used to filter out bad parts of the data, When a given time window has a standard deviation threshold, excess bad data are rejected. Data areas will be rejected if they exceed the standard deviation of the calibration data 20 times. Using this function from the command line, the time window is adjusted to delete bad data that might have been missed by ASR.(5)Since some data time series are longer, the EEG signal is down-sampled from 128 Hz to 100 Hz to effectively reduce the computational pressure on the system while ensuring the integrity of the EEG signal.

### 2.6. Statistical Analysis

SPSS 20.0 software was used for statistical analysis, means ± standard deviations were used for normally distributed data, continuous variables were represented by median (interquartile distance), and *t*-tests were used for group comparisons. If the distribution is not normal, Mann–Whitney u test was used. Repeated measures data were compared within groups using analysis of variance, count data comparisons were performed using the Chi-square test, and *p* < 0.05 was considered statistically significant.

## 3. Results

### 3.1. Difference of BIS between the Two Groups during Low Dose Propofol Induction

The BIS Reduction in the VS Group Was Significantly Greater Than that in the NVS Group when 0.5 mg/kg Propofol Anesthesia Was Induced, and the Difference Was Statistically Significant

The BIS data changes of the two groups of patients from 1 min to 5 min are shown in Table 3.

### 3.2. Forehead EEG Time–Frequency Graph and Power Graph

(1)Non-craniotomy, Non-Vegetative State Patient (NVS Group) EEG Time-Frequency Graph and Power Spectrum.(2)Traumatic Brain Injury VS Patient EEG Time-Frequency Graph and Power Spectrum.(3)Ischemic-Hypoxic VS Patient EEG Time-Frequency Graph and Power Spectrum.

According to the time–frequency spectrum, the energy around 10 Hz in the NVS group is prominent after anesthesia induction, while the energy at high frequencies is reduced (Figure 1B); VS patients with traumatic brain injury show a reduction in high-frequency energy and a concentration mainly below 10 Hz (Figure 2B). The α-band energy is prominent, and the 10–20 Hz band energy is higher, which is obviously different from the normal time-frequency characteristics (Figure 3B).

In the power spectrum, it can be seen that 5 min after anesthesia induction compared with 5 min before induction, the brain electrical energy of the NVS group is weakened, and from the change trend, a small wave peak, i.e., the α band peak, appears after 10 Hz (Figure 1D); the energy of traumatic brain injury VS patients is weakened, but no α wave peak appears (Figure 2D); in ischemic–hypoxic VS patients, there is no significant change in low-frequency energy, but high-frequency energy is significantly reduced, and a clear α wave peak can be observed slightly after 10 Hz (Figure 3D).

### 3.3. Changes in CRS-R and Clinical Outcome after 3 Months of SCS Surgery in 7 VS Patients

Three Months after SCS Treatment, we Followed up with the Patients in Collaboration with the Surgeon. Among them, one patient with traumatic brain injury was re-diagnosed from VS to MCS^−^, one ischemic–hypoxic patient’s CRS-R score increased by 1 point, and the remaining five VS patients had no changes in diagnosis or CRS-R score, as shown in Table 4.

## 4. Discussion

Patients in a VS are a distinct population characterized by impairments in consciousness, primarily resulting from traumatic brain injury, cerebrovascular diseases, or ischemic brain damage induced by cardiac arrest. The implementation of SCS surgery has emerged as a viable therapeutic approach for VS patients. This surgical procedure necessitates the patient to assume a prone position and undergo strict immobilization, thereby commonly necessitating the administration of general anesthesia. Due to the structural and functional damage to the brain in VS patients, the choice of anesthetic drugs and dosage and the monitoring of anesthesia depth have their own specialties. The goal is to not affect the expected outcome of the patient’s SCS postoperative results and to achieve better treatment and good outcomes. Currently, there are no more related reports domestically and internationally.

The anesthetic depth monitor, the BIS, is a derivative of statistical data obtained by NVS, mainly from β ratio, burst suppression ratio, and bispectral analysis of fast- and slow-wave relative synchronization [17,18,19,20]. Its effectiveness in monitoring the BIS in surgery anesthesia patients without consciousness disorders has been confirmed. The BIS holds significance in assessing the level of coma and predicting outcomes in individuals with severe craniocerebral injury [21]. However, there is a dearth of literature exploring the use of the BIS in monitoring anesthesia depth and providing precise guidance for clinical anesthesia in VS patients. In this study, the BIS was used to monitor the depth of anesthesia of patients. We used 1/2 of the recommended dose of propofol (1 mg/kg) in critically ill patients, which is 0.5 mg/kg, to induce general anesthesia and monitored the change of the BIS in patients from 1 to 5 min. It was found that the BIS of VS patients decreased significantly more than that of the NVS group when 0.5 mg/kg propofol was induced, and the depth of anesthesia required for normal (unconscious disorder) surgery was rapidly reached. Analysis of the reasons may be related to the potential recovery process, various secondary injuries, drug action, and self-metabolism of VS patients after brain damage, and it is an unstable state, resulting in reduced integration of the brain network [22]. Low-dose propofol can inhibit cerebral cortical electrical activity, which may be related to the weak cortical electrical activity of VS patients. Although 0.5 mg/kg of propofol given to VS patients can make their BIS value reach the depth of anesthesia required for normal-consciousness patients in the NVS group, the accuracy of the BIS in monitoring the depth of sedation in VS patients during surgery has not been strictly determined.

Electroencephalographic spectral analysis shows that after injecting 0.5 mg/kg of propofol for anesthesia induction, a prominent α frequency band around 10 Hz and decreased high-frequency energy can be seen in the NVS group of patients without consciousness disorders, which is consistent with the results reported by Purdon et al. [23]. When low-dose propofol was administered, α energy decreased, and γ and β energy increased in cortical circuits [24,25]. In the maintenance phase, propofol shows a combination of δ rhythm and α rhythm oscillations, and the α rhythm is highly correlated with the prefrontal lobe. The α rhythm is spatially coherent and restricts the communication of the frontal cortex circuit, thus mediating the unconscious state [26]. These dynamic EEG changes return to baseline again as propofol effects diminish and consciousness gradually returns [27]. Patients in a VS have reduced brain activity [28,29,30] in proportion to the severity of the injury [31]. Different VS patients show great differences in brain activity and EEG [32], and the δ power of VS patients is higher than that of NVS and MCS patients [33]. In nearly 80% of VS/UWS patients, the spectral power is dominated by the delta band [30,34], with no specific characteristics, which may be related to the structural changes of brain tissue. Jin et al. [35] found that patients in an ischemic anoxic vegetative state (VS) with a good prognosis showed an increase in α band power in the frontal lobe, which was similar to that of NVS. However, this change was not evident in patients with a poor prognosis. In this study, it was observed that patients in an ischemic hypoxic VS also showed an increase in α band power in the frontal region after receiving anesthesia (see Figure 3B). However, despite three months of observation, the patient’s consciousness did not improve significantly. This may be due to the fact that the observed recovery time is not long enough, as in some cases, the patient’s recovery may take longer and may even show significant improvement at a later stage. In addition, the use of propofol may have an effect on the EEG of patients in a vegetative state. Propofol inhibits harmonic generation in the brain, thereby reducing the overall activity level of the brain, manifested as an increase in alpha waves. This drug effect may be observed in electroencephalograms [36]. Arousal may also be related to the severity of brain tissue necrosis or brain cell damage in VS patients, which may interfere with the transmission of neurotransmitters and interfere with information transmission [37], thus affecting the recovery of consciousness in patients.

The EEG power spectrum of VS patients before anesthesia showed that the power of the slower band (δ,θ) increased, while the power of the higher band (α,β) decreased significantly [34,38]. In this study, it was found that under low-dose propofol anesthesia in traumatic VS patients, the high-frequency energy was observed to be weakened, and the main energy was concentrated below 10 Hz, mainly in the δ band, which was roughly similar to the results of Lechinger et al. [39,40,41]. The time–frequency pattern of patients in an ischemic anoxic VS is more special, with high energy in the 10–20 Hz frequency band and α and β waves as the main wave. Compared with NVS and patients with traumatic VS, the time–frequency characteristics are significantly different. The analysis may be due to the high metabolism and oxygen consumption of the cerebral cortex, the poor tolerance to hypoxia, and the cerebral cortex often presenting diffuse and multi-site damage involving the motor cortex and upper motor neurons.

In primates and normal humans, the GABA neurotransmitter promoter propofol is the most widely used anesthetic when studying brain mechanisms [42]. Propofol can inhibit or change neuronal activity [43]. There are few reports on whether propofol has significant effects on the fragile brain of VS patients, such as brain function, brain network connectivity, and brain network remodeling. Recent studies have found that propofol can predict the conscious outcome of VS patients [35], while there are few reports on whether propofol causes irreversible damage to the existing cerebral cortex, brain function, and brain network connectivity in VS patients. In view of the safety of VS patients, minimal impact, and better therapeutic outcomes, we used 1/2 of the recommended dose of propofol (1 mg/kg), 0.5 mg/kg for general anesthesia induction.

VS patients with different injury types have different changes in the response to propofol-induced loss of consciousness on EEG, and the complexity of EEG expression changes with the change of consciousness state, which may indirectly reflect brain activity and retained organizational capacity [44]. In our study, seven patients in a VS were followed up 3 months after surgery (the reasonable postoperative evaluation time recognized by the surgeon), and the CRS-R score was determined. It was found that the diagnosis of one patient with craniocerebral injury VS changed from VS to MCS-, the score of the patients with ischemia and hypoxia VS increased by 1 point, and the CRS-R score of the other five patients in a VS did not change significantly. The increase in the CRS-R may be related to SCS treatment. Judging from the overall CRS-R score, general anesthesia with a low dose of propofol does not seem to have a significant effect on VS patients’ awakening therapy, and a longer follow-up time is needed for evaluation in later studies.

Our study has several limitations. First, the sample size of each group was relatively small and needed to be validated in a larger cohort of patients, and there were also different types of patients with brain injury in the VS group. Secondly, there is no literature for reference on surgical anesthesia for such patients. Whether individualized adjustment of anesthesia regimen (drug selection of propofol and dose selection) has any impact on the long-term recovery of patients undergoing SCS surgery, especially whether the response of the fragile brain to propofol is correlated with long-term prognosis, needs further clinical research with a larger sample size. This can help VS patients attain better outcomes in clinical treatment. We will determine “whether propofol is helpful in restoring consciousness or has an ‘on/off’ effect” in further study.

## 5. Conclusions

In summary, patients in a VS have a consciousness disorder due to severe brain damage, anesthetic drugs have various degrees of influence on them, and the impact of their EEG is also different from that of NVS. We found that the increase in alpha waves in the EEG of VS patients with ischemia and hypoxia could not be used as an absolute criterion to judge the probability of consciousness recovery. Meanwhile, low-dose propofol may be a sedative choice for SCS implantation in VS patients.

## Figures and Tables

**Figure 1 brainsci-13-01608-f001:**
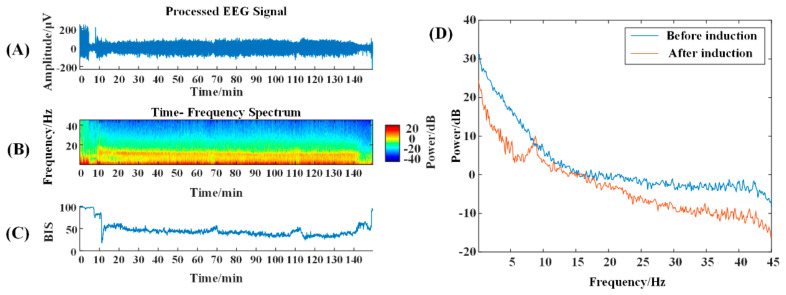
NVS Time-frequency graph and power spectrum. (**A**) The original EEG signal with a duration of 150 min; (**B**) The time-frequency spectrum over that time period; colors represent energy levels, with redder colors representing higher energy levels, and bluer colors representing lower energy levels. The BIS (range 0 to 100) can be seen in (**C**). The power spectrum graph can be seen in (**D**).

**Figure 2 brainsci-13-01608-f002:**
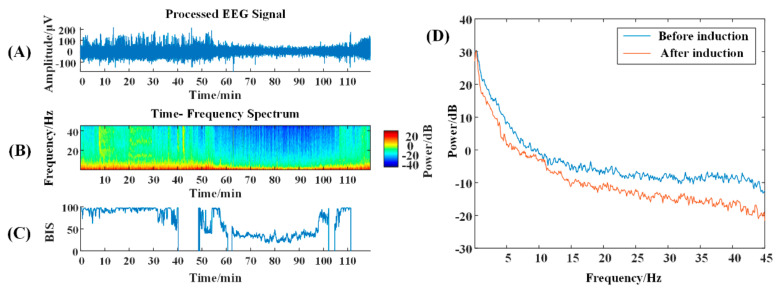
Time-frequency graph and power spectrum of traumatic-brain injury VS-patient. (**A**) The original EEG signal with a duration of 150 min; (**B**) The time-frequency spectrum over that time period; colors represent energy levels, with redder colors representing higher energy levels, and bluer colors representing lower energy levels. The BIS (range 0 to 100) can be seen in (**C**). The power spectrum graph can be seen in (**D**).

**Figure 3 brainsci-13-01608-f003:**
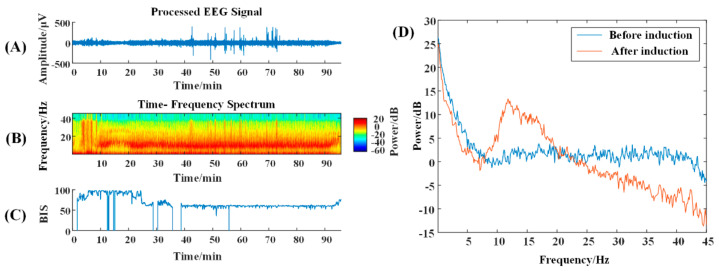
Time-frequency graph and power spectrum of ischemic-hypoxic VS patient. (**A**) The original EEG signal with a duration of 150 min; (**B**) The time-frequency spectrum over that time period; colors represent energy levels, with redder colors representing higher energy levels, and bluer colors representing lower energy levels. The BIS (range 0 to 100) can be seen in (**C**). The power spectrum graph can be seen in (**D**).

**Table 1 brainsci-13-01608-t001:** Comparison of general data of the two groups of patients.

Item	NVS Group (*n* = 5)	VS Group (*n* = 7)	*p*
Gender ratio (male/female)	2/3	4/3	1.000
Age (years)	41 (33–57)	41 (26–58)	0.745
Height (cm)	168 (160.5–171.5)	169 (163–172)	0.370
Weight (kg)	64 (57.5–70)	67 (60–72)	0.935
BIS value at room entry	97 (96.5–98)	89 (87–92)	0.004
Operation duration (min)	99 (89–118.5)	97 (89–116)	1.000

**Table 2 brainsci-13-01608-t002:** General data on 7 vegetative state patients.

Patient	Age/Sex	Distance from Doc (Months)	Etiology	CRS-R Total Score	Diagnosis	Saturation (%)
VS_1_	65/M	5	Traumatic	7	Hydrocephalus, L frontal Contusion, SAH	97
VS_2_	33/F	7	Traumatic	8	Bi frontal and temporal contusion, Left temporal haemorrhage	96
VS_3_	55/F	4	Anoxic	6	----	97
VS_4_	58/M	9	Traumatic	9	R temporal and frontal contusion, thalamus hemorrhage	95
VS_5_	21/M	2	Anoxic	8	----	97
VS_6_	41/F	5	Traumatic	8	Bifrontal and R occipital contusion, SAH	94
VS_7_	26/M	6	Traumatic	9	R temporal and frontal contusion, Right frontal haemorrhage	98

**Table 3 brainsci-13-01608-t003:** BIS values at different time points during 0.5 mg/kg propofol anesthesia induction in the two groups of patients ($\bar{X}$ ± s).

	0 min	1 min	2 min	3 min	4 min	5 min
VS Group	89.43 ± 2.64	48.57 ± 4.50	40.86 ± 2.12	42.57 ± 1.62	44.00 ± 1.63	45.71 ± 1.11
NVS Group	96.80 ± 0.84	89.80 ± 3.03	79.60 ± 3.05	81.00 ± 2.12	81.40 ± 2.41	82.20 ± 5.89
*t*-value	−6.925	−17.685	−26.143	−35.743	−32.261	−13.677
*p*-value	0.000	0.000	0.000	0.000	0.000	0.000

**Table 4 brainsci-13-01608-t004:** Changes in CRS-R scores before and 3 months after surgery in 7 VS patients.

	Preoperative CRS-R Score						3 Mouths Postoperative CRS-R Score					
Case	Hearing	Language	Vision	Communication	Motor	Arousal	Hearing	Language	Vision	Communication	Motor	Arousal
1	1	1	1	1	0	2	2	2	2	1	0	2
2	1	1	3	1	0	2	2	1	3	1	0	2
3	1	1	2	1	0	2	2	2	3	1	0	2
4	1	3	1	1	0	2	1	3	1	1	0	2
5	0	3	1	1	0	2	1	3	2	1	0	2
6	1	1	2	1	0	2	1	1	2	1	0	2
7	1	1	2	1	0	2	1	1	2	1	0	2

## Data Availability

Data available on request due to restrictions eg privacy or ethical. The data presented in this study are available on request from the corresponding author. The data are not publicly available due to: The data are currently shared with the department of Neurosurgery, and the data can be shared after publication.

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
