# Peer review of "EEG Changes during Propofol Anesthesia Induction in Vegetative State Patients Undergoing Spinal Cord Stimulation Implantation Surgery"

_brainsci, 2023, doi:10.3390/brainsci13111608_

Round 1

Reviewer 1 Report

Comments and Suggestions for Authors

This work shows how important to take into account the effect of  even low doses of propofol during the assestment of metabolism of 18FDG  in brain in UWS  patients using PET or studying the activity of neural networks on fMRI. 

It very interesting fact,  that  «patients with ischemic hypoxic UWS/VS showed an increase in α band power in the frontal region after receiving anesthesia» -  this indicates the presence of a «functional reserve» for the recovery consciousness in UWS/VS patients. However, you noted that despite 3 months of observation, the patient's consciousness did not improve significantly. It may be necessary to create  pharmacological conditions so that this restructuring of EEG lasts longer and then there will be a different outcome ... We are clearly dealing  with the «on-off» effect of pharmacological modulation of brain activity in DoC patients.  I advise authors to think about this in the future.

Author Response

Comments and Suggestions for Authors

This work shows how important to take into account the effect of  even low doses of propofol during the assestment of metabolism of 18FDG  in brain in UWS  patients using PET or studying the activity of neural networks on fMRI. 

It very interesting fact,  that  «patients with ischemic hypoxic UWS/VS showed an increase in α band power in the frontal region after receiving anesthesia» -  this indicates the presence of a «functional reserve» for the recovery consciousness in UWS/VS patients. However, you noted that despite 3 months of observation, the patient's consciousness did not improve significantly. It may be necessary to create  pharmacological conditions so that this restructuring of EEG lasts longer and then there will be a different outcome ... We are clearly dealing  with the «on-off» effect of pharmacological modulation of brain activity in DoC patients.  I advise authors to think about this in the future.

Author response #1

  • We agree with the reviewer and have done some research. As a short-acting drug, the effect of propofol on brain network remodeling in vegetative state patients has not been studied so far. Propofol is an ultra-short-acting sedative drug in clinical anesthesia, with a half-life of 1.8-8.3 minutes and a metabolite excretion time of 34-60 minutes. After searching studies, we found that there is basically no research on the time of brain network reconstruction with propofol. It is possible that propofol affects its reconstruction, as your opinion, leading to impaired recovery after 3 months. This gives us a good suggestion, and we will add this aspect in future research (including CRS-R assessment combined with fMRI). To investigate the effect of propofol on brain remodeling in such patients, and to obtain the best follow-up time.

  • About the “on/off” effect. We agree with the opinion of reviewer and have added it in the limitation part. Based on the comments, we will add “whether propofol is helpful in restoring consciousness or has an "on/off" effect” into the further study.

Reviewer 2 Report

Comments and Suggestions for Authors

The submitted article evaluates the changes in native EEG, EEG spectrogram, and BIS EEG monitoring when administering low-dose propofol to patients with chronic consciousness disorders (vegetative state -VS). 

This article has considerable relevance and interest to a wide audience, as there are no clear standards for anaesthesia in these patients. 

I have the following concerns about the article:

Abstract: The conclusion within the abstract does not correspond to the material and methods. It is not clear what this sentence means«This difference may be the fragile brain's own sexual protection, which may also be the basis of wakefulness therapy» 

  1.  

  1. Methods:  The authors acquired EEG from a single channel over the left hemisphere regardless of lesion location, although there were 2 patients with traumatic lesions of the left hemisphere in the cohort studied. This may have impacted the results, leading to a false-positive reduction in EEG amplitude, especially as there was no comparison with the contralateral hemisphere. 

  2.  
  3. The authors used the t-test to compare the groups, but with such a small number of patients, non-parametric methods of statistical analysis (median and interquartile interval) should be used. 

  4.  
  5. The authors report a difference in the EEG spectrogram between the traumatic and anoxic groupsThey only support their opinion in Figures 2 and 3 with 1 patient from the traumatic group and 1 patient from the anoxic group. However, it is necessary to summarise the data of the remaining patients from the traumatic and anoxic VS groups. 

  6.  
  7. The reasons for the use of general anaesthesia in the NVS group are not specified in detail. A table showing the conditions for which general anaesthesia was given and whether any of the patients had cerebral pathology is required. 

  8.  
  9. Table 4 has numerous errors and needs to be redesigned and reformatted. 

  10.  
  11. The submitted manuscript lacks a conclusion and a summary 

Comments on the Quality of English Language

The English language requires improvement and correction, in particular terms such as "normal group", "normal person" etc. should be avoided.

Author Response

Comments and Suggestions for Authors

The submitted article evaluates the changes in native EEG, EEG spectrogram, and BIS EEG monitoring when administering low-dose propofol to patients with chronic consciousness disorders (vegetative state -VS). 

This article has considerable relevance and interest to a wide audience, as there are no clear standards for anaesthesia in these patients. 

I have the following concerns about the article:

  1. Abstract: The conclusion within the abstract does not correspond to the material and methods. It is not clear what this sentence means: «This difference may be the fragile brain's own sexual protection, which may also be the basis of wakefulness therapy»

Author response #1

Sorry for the inconvenient. We have revised the conclusion in the Abstract section (Page 1, Line 37-39).

And about the confusing sentence «This difference may be the fragile brain's own sexual protection, which may also be the basis of wakefulness therapy», we are sorry for the description before. About this, we want to express that VS patients with different injury types will have different EEG manifestations under low dose of propofol (traumatic VS patients: high-frequency capacity is reduced and mainly concentrated below 10Hz; VS patients with ischemia and hypoxia: α band energy is prominent and 10-20Hz frequency band energy is high), which may indicate that damaged nerve cells in such patients have partial function, and the probability of consciousness recovery after SCS treatment is higher. We have rewritten discussion section and this sentence has been deleted.

  1. Methods:  The authors acquired EEG from a single channel over the left hemisphere regardless of lesion location, although there were 2 patients with traumatic lesions of the left hemisphere in the cohort studied. This may have impacted the results, leading to a false-positive reduction in EEG amplitude, especially as there was no comparison with the contralateral hemisphere. 

Author response #2

Sorry for the confusion. BIS monitors the forehead and is not affected by trauma and does not affect the surgical area. In future studies, we will pay attention to this problem and change BIS to 19-lead EEG for relevant EEG monitoring to obtain more specific data in brain regions.

3.The authors used the t-test to compare the groups, but with such a small number of patients, non-parametric methods of statistical analysis (median and interquartile interval) should be used. 

Author response #3

We agree with the reviewer and according to suggestions, we will use non-parametric test to calculate the relevant data and fill in the table. The revised table and corresponding statistical methods are also modified in the paper (Page3, Line 99; P5, Line 183-187)

4.The authors report a difference in the EEG spectrogram between the traumatic and anoxic groups. They only support their opinion in Figures 2 and 3 with 1 patient from the traumatic group and 1 patient from the anoxic group. However, it is necessary to summarise the data of the remaining patients from the traumatic and anoxic VS groups.

Author response #4

Yes, we agreed with the reviewer’s opinion. The EEG spectrum involved in this study represents the EEG changes of traumatic VS patients and ischemic hypoxia VS patients under the effect of low-dose propofol. After discussing with the EEG experts, they suggested that more representative patients should be selected so we can explain more specifically. Therefore, we selected the brain time spectrum and power spectrum of 1 traumatic VS patient and 1 ischemic hypoxia patient as representatives, and the EEG of other patients was basically similar.

5.The reasons for the use of general anaesthesia in the NVS group are not specified in detail. A table showing the conditions for which general anaesthesia was given and whether any of the patients had cerebral pathology is required.

Author response #5

Sorry we didn't elaborate on this before. Due to ethical and other limitations, it is difficult to enroll volunteers into this study. We choose gynecological laparoscopic surgery patients (general anesthesia) and exclude history of brain disease. So, it is easier to obtain relevant data in patients with vegetative state at the same time.

6.Table 4 has numerous errors and needs to be redesigned and reformatted. 

Author response #6

We agree with reviewer's opinion and we have modified the contents of the form thoroughly.

Thank you for pointing out these mistakes. Other forms has been further reviewed to ensure the accuracy of the forms and the content of this study.

  1. The submitted manuscript lacks a conclusion and a summary.  

Author response #7

Sorry for the inconvenient, we have revised the manuscript and highlighted the conclusion part in discussion section.(Page 10,Line 347)

  1. Comments on the Quality of English Language

The English language requires improvement and correction, in particular terms such as "normal group", "normal person" etc. should be avoided.

Author response #8

We agree with the reviewer and the manuscript have been checked by a native English speaker.

Reviewer 3 Report

Comments and Suggestions for Authors

Overall, this is an excellent small group design that assesses the effect of Propofol on the BIS and coma-related outcomes of 7 vegetative state patients and 5 control patients as the VS patients underwent spinal stimulator implantation. 

The methods are sound and the results are interesting and informative. Unfortunately, the stimulator treatment did not results in recovery for the VS patients, but the EEG collected by the BIS machine yielded some interesting results. The increased alpha (10-20 Hz) in the I-H patient is suggestive of an interesting EEG signature for this type of patient during anesthesia.

Line 69: Please define BIS for the reader, as well as what it is comprised of and how it is calculated.

In addition to reference 13, additional literature review is required to frame the place of this study within the literature. Please strengthen the introduction with a review of relevant studies that have found mixed results on low dosage propofol release.

I see on Line 325 you address the lack of baseline studies. I did find one study that would be valuable for you to include in the introduction. "Jaggi, P., Schwabe, M. J., Gill, K., & Horowitz, I. N. (2003). Use of an anesthesia cerebral monitor bispectral index to assess burst-suppression in pentobarbital coma. Pediatric neurology28(3), 219-222."

Please clarify the ethical and consent process for the normal and VS patients.

Comments on the Quality of English Language

An example of a small English problem is: "the patient needs to be prone position."

The correct phrase is "the patient needs to be in a prone position."

Please proofread for these small errors.

Author Response

Comments and Suggestions for Authors

Overall, this is an excellent small group design that assesses the effect of Propofol on the BIS and coma-related outcomes of 7 vegetative state patients and 5 control patients as the VS patients underwent spinal stimulator implantation. 

The methods are sound and the results are interesting and informative. Unfortunately, the stimulator treatment did not results in recovery for the VS patients, but the EEG collected by the BIS machine yielded some interesting results. The increased alpha (10-20 Hz) in the I-H patient is suggestive of an interesting EEG signature for this type of patient during anesthesia.

  1. Line 69: Please define BIS for the reader, as well as what it is comprised of and how it is calculated.

Author response #1

We agree with the reviewer and added the define of BIS in Page2 ,Line73-79. As follows:BIS refers to the determination of linear components of EEG (frequency and power), analysis of the nonlinear relationship between component waves (phase and harmonic), the selection of various EEG signals representing different sedation levels, standardization and digital processing, is a more accurate method to judge the sedation level and monitor the depth of anesthesia. It calculates a single dimensionless variable between 0 and 100. A patient with an isoelectric EEG has a score of 0, while if fully awake should have a BIS score close to 100. Bis scores of 40 to 55 are typical during general anesthesia.

2.In addition to reference 13, additional literature review is required to frame the place of this study within the literature. Please strengthen the introduction with a review of relevant studies that have found mixed results on low dosage propofol release.

Author response #2

We have found some relevant studies and cited in the introduction part, thank you for your suggestion. (Page 2, Line68).

  • I see on Line 325 you address the lack of baseline studies. I did find one study that would be valuable for you to include in the introduction. "Jaggi, P., Schwabe, M. J., Gill, K., & Horowitz, I. N. (2003). Use of an anesthesia cerebral monitor bispectral index to assess burst-suppression in pentobarbital coma. Pediatric neurology28(3), 219-222."

Author response #3

Thank you for your recommendation, we have studied thoroughly and inserted into the study (Page2, Line72).

  1. Please clarify the ethical and consent process for the normal and VS patients.

Author response #4

This study has passed the ethical application, and for NVS patients (gynecological operation), we will sign the informed consent form with them. For VS patients, since their special conditions (disabled), informed consent was signed by their agent (legal representative).

  1. An example of a small English problem is: "the patient needs to be prone position."

The correct phrase is "the patient needs to be in a prone position."

Please proofread for these small errors.

Author response #5

Sorry for the inconvenient. We have corrected the phrases in this article, and the manuscript have been checked by a native English speaker. Thank you for your understanding

Round 2

Reviewer 2 Report

Comments and Suggestions for Authors

The authors have corrected and replied to all the comments I've made. The manuscript can be published in its present form.